DATA RELEASE

# Whole-genome re-sequencing of the Baikal seal and other phocid seals for a glimpse into their genetic diversity, demographic history, and phylogeny

Marcel Nebenführ[1,2,3,*], Ulfur Arnason[4] and Axel Janke[1,2,3]

1 Institute for Ecology, Evolution and Diversity, Goethe University, Max-von-Laue-Strasse. 9, Frankfurt am Main, 60438, Germany

2 LOEWE-Centre for Translational Biodiversity Genomics (TBG), Senckenberg Nature Research Society, Georg-Voigt-Straße 14-16, Frankfurt am Main, 60325, Germany

3 Senckenberg Biodiversity and Climate Research Centre (BiK-F), Georg-Voigt-Straße 14-16, Frankfurt am Main, 60325, Germany

4 Department of Neurosurgery, Skane University Hospital in Lund, 222 42, Lund, Sweden

## ABSTRACT

The Baikal seal (*Pusa sibirica*) is a freshwater seal endemic to Lake Baikal, where it became landlocked million years ago. It is an abundant species of least concern despite the limited habitat. Research on its genetic diversity had only been done on mitochondrial genes, restriction fragment analyses, and microsatellites, before its reference genome was published. Here, we report the genome sequences of six Baikal seals, and one individual of the Caspian, ringed, and harbor seal, re-sequenced from Illumina paired-end short read data. Heterozygosity calculations of the six newly sequenced individuals are similar to previously reported genomes. Also, the novel genome data of the other species contributed to a more complete phocid seal phylogeny based on whole-genome data. Despite the isolation of the land-locked Baikal seal, its genetic diversity is comparable to that of other seal species. Future targeted genome studies need to explore the genomic diversity throughout their distribution.

**Submitted:** 29 July 2024

\* Corresponding author. E-mail: marcel.nebenfuehr@senckenberg.de

Preprint submitted at https://doi.org/10.1101/2024.10.19.619210

**Subjects** Animal and Plant Sciences, Genetics and Genomics, Ecology

# DATA DESCRIPTION

## Background information

With a depth of 1,637 m, and an age of 25 million years, Lake Baikal is the world's deepest and one of the oldest freshwater lakes [1, 2]. Due to the extreme diversity of landscapes surrounding the 636 km long and 80 km wide crescent-shaped waterbody, Lake Baikal harbors a highly diverse flora and fauna, most notably the iconic Baikal seal (*Pusa sibirica*) [3].

The Baikal seal is a large carnivorous mammal, reaching 165 cm in length and weighing up to 130 kg. The fact that it is found only in this landlocked, isolated habitat makes it unique among all pinnipeds, representing the only freshwater seal [3, 4]. Apart from its silver-gray back and yellowish-white belly that characterizes adults, the Baikal seal is well adapted to its unique habitat under isolated conditions. One adaptation common to all

pagophilic, sea-ice-preferring Arctic phocids is their much larger claws, needed to maintain breathing holes in the ice [5]. However, Baikal seals face hard freshwater ice and have evolved even stronger claws, followed by those of the ringed seal (*Pusa hispida*), which inhabits softer and less dense sea ice. The third species of the genus *Pusa* - the Caspian seal (*Pusa caspica*) occurs in areas where ice is thin or entirely absent, and consequently has developed relatively weak claws by comparison [6].

Even though the *Phoca/Pusa* group harbors iconic species, it remains understudied on a genome-wide scale. To date, only one study of the Baikal seal on genomic data is available, as its genome was recently sequenced for a first glimpse into its genomic structure, demographic history, and a first phylogeny of the *Phoca/Pusa* group on genomic data [7]. Meanwhile, these phylogenetic relationships are not completely resolved, as several taxa have not been included in the tree yet.

To study the genomic diversity of the Baikal seal using more solid data and to contribute baseline data to the limited genomic data available for seals, we re-sequenced the genomes of six Baikal seal individuals (NCBI:txid9719) to a 30-fold coverage using Illumina short-read technology. Further, we sequenced one individual of the Caspian seal (NCBI:txid693431), the harbor seal (*Phoca vitulina*, NCBI:txid9720), and the ringed seal (NCBI:txid9718). We mapped all reads against the chromosome-level reference genome of the Baikal seal to generate reference-based assemblies for assessing the seal's genetic diversity and demographic history. In addition, including our newly sequenced seal species, we present an expanded phylogeny of the phocid seals based on genome-wide data.

## Sampling, DNA extraction, and sequencing

Heart and liver biopsy samples from two male and four female Baikal seals were collected following local ethical laws and regulations and provided by the Eastern Siberian Scientific and Fishery Production Centre (VOSTSIBRYBCENTR), Ulan-Ude, Russia. The tissue was preserved in ethanol for shipment and then stored at −80 °C. Unfortunately, no detailed geographic or individual records were kept for the 30-year-old samples. High molecular weight DNA (hmwDNA) was extracted from the samples using the QIAGEN DNeasy Blood and Tissue Kit. The quantity and quality of the DNA was assessed using the Genomic DNA ScreenTape on the Agilent 2200 TapeStation system (Agilent Technologies) (RRID:SCR_014994). The geographic origin of the harbor seal, Caspian seal, and ringed seal hmwDNA-sample was also not recorded, but it was identical to the DNA used in [8]. DNA samples from all individuals were sent to BGI Genomics (Shenzhen, China) to generate 150 PE ILLUMINA libraries.

## Reference-based assemblies

Sequencing reads were trimmed and filtered using Trimmomatic v0.32 (RRID:SCR_011848) [9] with the following settings: *ILLUMINACLIP:TruSeq3-PE-2.fa:2:30:10 SLIDINGWINDOW:4:20 MINLEN:40 TOPHRED33*. Subsequently, clean reads were mapped against the available Baikal seal reference genome using BWA v0.7.17-r1188 r1188 (RRID:SCR_010910) [10] and Samtools v1.15 [11]. Duplicated reads were removed from all files using Picard MarkDuplicates v.3.1.1 (RRID:SCR_006525) [12]. All files were filtered for mapping quality and alignment score using Samtools view with the following settings: *-bhq 20 -f 0x2 -F4 -e '[AS] >=100'*. The quality of the final files was assessed using Qualimap v.2.2.2-dev (RRID:SCR_001209) [13].

**Table 1.** Mapping statistics of all reference-based assemblies, including publicly available individuals. Mapping statistics of the reference-based assemblies of the newly sequenced seals, together with the short-read data taken from databases estimated with Qualimap. All reads were mapped against the Baikal seal reference genome.

| Sample (Accession) | Mapped reads (no/%) | Average mapping quality | Error rate | Average coverage |
|---|---|---|---|---|
| Ringed seal-2 (*Pusa hispida*) | 268,582,518 (100%) | 59.7 | 0.0067 | 17.3 |
| Ringed seal-1 (*Pusa hispida*) (ERR10317392) | 83,660,001 (100%) | 59.7 | 0.0077 | 5.3 |
| Harbor seal (*Phoca vitulina*) | 317,265,451 (100%) | 59.7 | 0.0074 | 20.5 |
| Grey seal (*Halichoerus grypus*) (SRR16086822) | 200,683,394 (100%) | 59.6 | 0.0073 | 12.1 |
| Spotted seal (*Phoca largha*) (SRR6433060) | 977,600,301 (100%) | 59.7 | 0.0073 | 63.3 |
| Caspian seal (*Pusa caspica*) | 246,366,449 (100%) | 52.2 | 0.0296 | 4.7 |
| Bearded seal (*Erignathus barbatus*) (SRR12437603) | 448,728,409 (100%) | 59.6 | 0.0185 | 29.3 |
| Baikal seal-1 (*Pusa sibirica*) | 500,860,831 (100 %) | 59.6 | 0.007 | 32.4 |
| Baikal seal-2 (*Pusa sibirica*) | 508,032,439 (100%) | 59.6 | 0.0066 | 32.9 |
| Baikal seal-3 (*Pusa sibirica*) | 503,314,156 (100%) | 59.6 | 0.007 | 32.6 |
| Baikal seal-4 (*Pusa sibirica*) | 472,340,010 (100%) | 59.7 | 0.0069 | 30.6 |
| Baikal seal-5 (*Pusa sibirica*) | 505,505,651 (100%) | 59.7 | 0.0067 | 32.8 |
| Baikal seal-6 (*Pusa sibirica*) | 473,515,735 (100%) | 59.6 | 0.0037 | 30.6 |

From all six Baikal seal individuals, 486–522 million short reads were sequenced, yielding an average of 32-fold coverage per individual. Sequencing of the harbor seal yielded a total of 317 million short reads, resulting in a 20-fold coverage. From the Caspian seal, 246 million short reads were sequenced, resulting in a 4.7-fold coverage. Sequencing of the ringed seal yielded a total of 268 million short reads, resulting in a 17.3-fold coverage (Table 1).

## Demographic history

The historical effective population sizes ($N_e$) of the Baikal seals, the ringed seal, and the harbor seal were estimated using Pairwise Sequentially Markovian Coalescent (PSMC) v0.6.5-r67 (RRID:SCR_017229) [14]. To this end, consensus genome sequences were generated from all reference-based assemblies using BCFtools v1.14 (RRID:SCR_005227) [11]. To account for a bias due to uneven sequencing depths, the input bam files were down-sampled to a depth of 17× using samtools view. The resulting files were filtered to remove sites with mapping quality <30 and read depth <10 or above twice their mean depth. The PSMC analysis was run with 25 iterations and 100 rounds of bootstrapping using the default atomic interval set for humans. To scale the analysis, a mutation rate of $0.7 \times 10^{-8}$ substitutions per nucleotide and generation was chosen [15], and generation times for each species were obtained from IUCN [16] being 21.6 years for the Baikal seal, 18.6 years for the ringed seal, and 14.8 years for the harbor seal.



Our PSMC analysis of the Baikal seals showed an increase in individuals from ~5–2 Ma ago, where a population maximum was reached. Afterwards, the plot shows a steep population decline until ~800 ka ago. Then, the $N_e$ increased again until ~300 ka ago, from when it shows a stable course until ~20 ka ago. From ~20–10 ka ago, the analysis shows another steep decline, reaching a population minimum lasting until present days. The analysis is congruent with previous PSMC results of other Baikal seal individuals, indicating a shared population size history of all individuals, and facilitates the hypothesis of a stabilization of the Baikal seal's $N_e$ due to balanced and stable habitat conditions after colonizing lake Baikal [7].

For the ringed seal, PSMC reconstructed a similar demographic history compared to the Baikal seals from ~5–1.5 Ma ago. Contrary to the Baikal seal, the $N_e$ of the ringed seal shows no decline afterwards but an increase until ~250 ka ago, and a population maximum from 150–100 ka ago. In general, the ringed seal shows the highest $N_e$ estimates among all analyzed pinnipeds, which is expected, as it is the most abundant arctic marine mammal [17]. From ~5 Ma ago until ~2 Ma ago, the ringed seal's and the Baikal seal's demographic histories show shared trajectories. The time at ~2 Ma ago, where the two species' plots merge, may mark the period in which they diverged. Based on divergence time estimates, this split is assumed to have happened ~1.15–1.7 Ma ago [18, 19].

Finally, our PSMC analysis of the harbor seal reports a population maximum at ~2Ma ago. After a small decline at ~1.5 Ma ago, its $N_e$ remained stable until ~300 ka ago. From that time, its $N_e$ declined until ~100 ka ago and stabilized again until 30 ka ago. Afterwards, the $N_e$ began to decrease again.

## Genome-wide heterozygosity

To assess the genetic diversity of the seals, genotype likelihoods and calls were generated from the reference-based assemblies using the bcftools v1.12 (RRID:SCR_005227) mpileup and call pipeline [11] following the BAM2VCF_run_mpileup_parallel_HIGHWAY script of the Fastq2VCF pipeline [20, 21] with default settings. During genotype calling, the "group-samples" option was used to assign each individual to its unique group and disable the option of influencing genotype calls based on information from other samples. Sites with a read depth below three were masked using the bcftools filter pipeline. For all individuals combined, sites with a minimum read depth of 150 (resembling 6× the number of samples) and a maximum depth of 700 (resembling 1.5 times the mean depth of the samples) were retained.

Subsequently, the custom-built Darwindow tool [22] based on Tabix v1.12 [23] was used to count the number of retained homozygous and heterozygous sites per sample on a sliding-window basis, using non-overlapping windows with a fixed size of 20 kb. Next, genomic regions were extracted and converted into heterozygosity (He) estimates.

A subset of biallelic sites was extracted from scaffolds for genomic variant analyses by filtering on levels of missing data. As a result, 1,166,730 single nucleotide variants were retained. The data was then converted from .vcf format into PED/RAW and MAP/BIM with the help of Plink v1.90b6.21 (RRID:SCR_001757) [24] (using the flags *make-bed, recode A, chr-set 95, and allow-extra-chr*). Analyses were performed in RStudio v2024.09.1 [25] with R v4.4.1 [26] using wrapper functions of the R package SambaR [27]. The data was imported

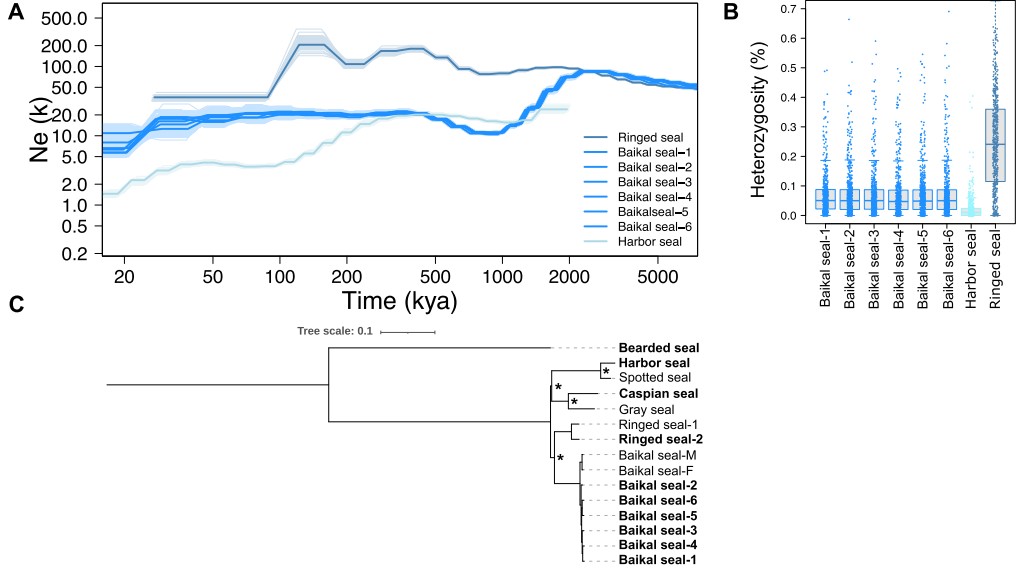

**Figure 1.** (A) Estimated $N_e$ of the Baikal seals using PSMC analysis. The *x*-axis shows time, and the *y*-axis shows $N_e$. Plots were scaled using a mutation rate (μ) of $0.7 \times 10^{-8}$ substitutions nucleotide$^{-1}$ generation$^{-1}$, and generation times for each species taken from IUCN [19]. (B) Genome-wide heterozygosity estimated from genomic 20-kbp windows. (C) Maximum-likelihood phylogeny, including the newly sequenced seal species (indicated in bold), estimated from 1,166,730 genome-wide single-nucleotide variants. The scale bar shows nucleotide substitutions per site, and black asterisks indicate full bootstrap support. The tree was arbitrarily rooted at the bearded seal.

into R and stored in a genlight object using the function "read.PLINK", which is part of the R package adegenet-2.1.1 [28]. The data was then filtered using SambaR's "filterdata" function with *indmiss=0.1, snpmiss=0.01, min_mac=2, and dohefilter=TRUE.*

Sliding-window heterozygosity calculations of the newly sequenced seals are shown in Figure 1B. The results show an average value of 0.066% for the Baikal seals (BS-1: 0.067%, BS-2: 0.066%, BS-3: 0.066%, BS-4 0.065%, BS-5: 0.066%, and BS-6: 0.066%), which is similar to previous genome-wide heterozygosity calculations for two Baikal seals that show values of 0.61 hetSNPs/kbp (0.061%) and 0.66 hetSNPs/kbp (0.066%) [7]. Further, with 0.021%, estimates were the lowest for the harbor seal corresponding to previous genome-wide He estimates of this species [29].

The estimated He of 0.24% for the ringed seal is among the highest for all pinniped species based on genomic data [7]).

## Phylogeny

To reconstruct the relationships among the seven seal species, a phylogenomic analysis was done on variable genomic sites across species. The program vcf2phylip.py v2.8 [30] was used for conversion into the phylip format. Next, a maximum likelihood phylogeny was computed using IQ-Tree v2.1.4-beta (RRID:SCR_017254). The analysis was run with 1,000 replicates for bootstrap analysis and the Shimodaira–Hasegawa approximate likelihood ratio test, respectively. The analysis based on 1,166,730 genomic single-nucleotide variants reconstructed the phylogenetic relationships shown in Figure 1C. The tree was arbitrarily rooted at the bearded seal. Together with the ringed seal as sister lineage, the Baikal seals are separated from the remaining phocid species. The six newly sequenced and the two published individuals of the Baikal seal fall together on a shared branch without much

variation between them. Further, the tree shows that the harbor seal falls together with the spotted seal, and the Caspian seal with the gray seal. In concordance with other studies based on mitochondrial genes, nuclear genes, or a combined dataset, the tree shows the spotted seal and the harbor seal forming one separated clade [8, 18, 31]. Additionally, it supports previous findings based on 50 maximum likelihood gene trees depicting the Caspian seal and the gray seal forming one separated clade [32]. The same is shown by full mitochondrial sequences, even though support values are low [8]. Our results indicate that the Baikal seal is the closest relative of the ringed seal, with full support, as confirmed by Restriction Analysis of mtDNA fragments [33] and the Bayesian phylogeny based on a combined nt/mtDNA dataset, although BPP values are poor. Moreover, the close relationship between the Baikal seal and the ringed seal is aided by skull measurements [34]. Meanwhile, the internal phylogeny of our tree with the basal split within the *Phoca*/*Pusa* group separating the Baikal seal and ringed seal from the remaining species forming sister clades with Caspian/gray seal on the one hand, and the harbor/spotted seal on the other hand, is in contradiction to any other previous phylogenetic study of this group. While previous research based on less than full mitochondrial sequences was not even able to reconstruct dichotomous trees of the *Phoca*/*Pusa* group [35], studies based on different tree-building methods of whole mitochondrial sequences, a few nuclear loci, or both were also incapable of resolving the phylogeny unambiguously. Hence, we conclude that those datasets were insufficient for resolving the actual phylogeny of this group. Unlike other available studies on the phylogenetic relationships of the *Phoca*/*Pusa* group that so far have been inferred by very limited data, we present a phylogeny on the most comprehensive dataset to date. For the first time, the dataset based on more than one million of genomic variants could resolve the phylogeny of the *Phoca*/*Pusa* group with full bootstrap support including splits within that evolutionary group that have been problematic.

## CONCLUSION

Here, we report reference-based assemblies of six newly sequenced Baikal seal individuals, one individual of the ringed seal, as well as the first short-read data of the harbor seal and the Caspian seal. Unfortunately, the low quality of the DNA from the Caspian seal allowed only to produce short-read sequences and limited the evolutionary analyses to single-nucleotide variants. Nevertheless, the novel data are a valuable addition to the growing database of seal genomes. With our newly generated sequencing data, we were not only able to extend the phylogeny of the *Phoca*/*Pusa* group on genome-wide data but could broaden the view into the genetic structure and diversity of the Baikal seal. Hopefully, this new baseline data will be the cornerstone for larger population genomics studies on the world's only freshwater pinniped.

## DATA AVAILABILITY

The raw sequencing files of all new pinniped samples were uploaded to NCBI under the BioPoject PRJNA1142506, BioSamples SAMN42945286–SAMN42945294. All other data, including the repeat and gene annotation, was uploaded to the GigaDB repository [36], with separate entries for the individual species genomes [37–40].

## ABBREVIATIONS

He, heterozygosity; hmwDNA, high molecular weight DNA; $N_e$, historical effective population size; PSMC, Pairwise Sequentially Markovian Coalescent.

## DECLARATIONS

### Ethics approval and consent to participate

The authors declare that ethical approval was not required for this type of research.

### Competing interests

The authors declare that they have no competing interests.

### Authors' contributions

See the Author Contributorship tabs for details. All authors read and approved the final manuscript before submission.

### Funding

The present study is a result of the Centre for Translational Biodiversity Genomics (LOEWE-TBG) and was supported through the program 'LOEWE-Landes-Offensive zur Entwicklung Wissenschaftlich-ökonomischer Exzellenz' of Hesse's Ministry of Higher Education, Research, and the Arts. No additional external funding was used.

### Acknowledgements

We are grateful for the seal samples Evgeny Petrov (The Eastern Siberian Scientific and Fishery Production Centre) provided in 1995 for the Arnason et al. (2005) paper. Unfortunately, we could not get in contact with him for co-authorship.

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
