## [Reviewer Report]

Indicate in the comments box below whether you are happy with the changes made or if the manuscript is unacceptable.Comments on revised manuscriptThe authors carefully addressed most of my concerns. Although I still doubt about the mapping rate (I did no find the mapping report attached), I am happy to accept this manuscript.Indicate in the comments box below whether you are happy with the changes made or if the manuscript is unacceptable.Comments on revised manuscriptThe authors carefully addressed most of my concerns. Although I still doubt about the mapping rate (I did no find the mapping report attached), I am happy to accept this manuscript.

---

## [Editor Report]

Editor’s AssessmentDue to them being found in the landlocked, isolated habitat of Lake Baikal makes the Baikal Seal (Pusa sibirica) unique among all pinnipeds as the only freshwater seal. This paper presents reference-based assemblies of six newly sequenced Baikal seal individuals, one individual of the ringed seal, as well as the first short-read data of the harbor seal and the Caspian seal . This data aiding the study of the genomic diversity of the Baikal seal and to contribute baseline data to the limited genomic data available for seals. Peer review extended the description of the used tools and parameters in the revised manuscript, and provided some more information on the methods..This newly generated sequencing data hopefully now helps to extend the phylogeny of the Phoca/Pusa group on genome-wide data and can also broaden the view into the genetic structure and diversity of the Baikal seal.Editor’s AssessmentDue to them being found in the landlocked, isolated habitat of Lake Baikal makes the Baikal Seal (Pusa sibirica) unique among all pinnipeds as the only freshwater seal. This paper presents reference-based assemblies of six newly sequenced Baikal seal individuals, one individual of the ringed seal, as well as the first short-read data of the harbor seal and the Caspian seal . This data aiding the study of the genomic diversity of the Baikal seal and to contribute baseline data to the limited genomic data available for seals. Peer review extended the description of the used tools and parameters in the revised manuscript, and provided some more information on the methods..This newly generated sequencing data hopefully now helps to extend the phylogeny of the Phoca/Pusa group on genome-wide data and can also broaden the view into the genetic structure and diversity of the Baikal seal.

---

## [Reviewer Report]

Reviewer name and names of any other individual's who aided in reviewer Yaolei ZhangDo you understand and agree to our policy of having open and named reviews, and having your review included with the published papers. (If no, please inform the editor that you cannot review this manuscript.)YesIs the language of sufficient quality?YesPlease add additional comments on language quality to clarify if needed
Are all data available and do they match the descriptions in the paper? NoAdditional CommentsAre the data and metadata consistent with relevant minimum information or reporting standards? See GigaDB checklists for examples <a href="http://gigadb.org/site/guide" target="_blank">http://gigadb.org/site/guide</a>NoAdditional CommentsIs the data acquisition clear, complete and methodologically sound?NoAdditional CommentsIs there sufficient detail in the methods and data-processing steps to allow reproduction?NoAdditional CommentsIs there sufficient data validation and statistical analyses of data quality? NoAdditional CommentsIs the validation suitable for this type of data?NoAdditional CommentsIs there sufficient information for others to reuse this dataset or integrate it with other data?NoAdditional CommentsAny Additional Overall Comments to the AuthorOverall, the newly generated data from this study are valuable, but the authors have not effectively analyzed and interpreted the data. The entire paper appears to be more like an undergraduate bioinformatics homework exercise, with the results resembling a middle school student's description of a picture. Additionally, there are several major issues: 1. Background investigation is not sufficient: Genomic data on the Baikal seal has been publicly available five years ago, including a chromosome-level genome assemby with much higher quality, such as contig N50, which is nearly ten times higher than the reference genome used by the author in this study. 2. Methodology is unclear: The description of the software and parameters used is incomplete. A proper methodological description should allow a basic bioinformatics analyst to quickly reproduce the results of the paper. However, with the current description, there are too many missing details in the methodology section. 3. Data issues: • a. For publicly available data, the authors did not provide detailed descriptions of the accession numbers. • b. For the newly generated data in this study, the author did not sufficiently describe the data quality to support their conclusions. • c. In the supplementary table, the author show 100% mapping rates of sequencing reads for all samples. Having worked on numerous resequencing projects, I have rarely encountered 100% mapping rates, especially when aligning to different species. The author should check this. 4. Basic analytical skill/experience is lacking: For example, the PSMC analysis, sequencing depth can directly affect the results, but the author did not consider this issue and proceeded to compare curves generated from different sequencing depths directly. Additionally, how was the mutation rate (μ) derived? The generation time is only mentioned as coming from IUCN, but values are not provided in the paper. Moreover, in the genetic diversity section, is calculating heterozygosity only sufficient to be considered a measure of genetic diversity? Hope the author to read some re-sequencing papers thoroughly.RecommendationReject (Unsound or Unusuable)

---

## [Reviewer Report]

Reviewer name and names of any other individual's who aided in reviewer Stephen GaughranDo you understand and agree to our policy of having open and named reviews, and having your review included with the published papers. (If no, please inform the editor that you cannot review this manuscript.)YesIs the language of sufficient quality?YesPlease add additional comments on language quality to clarify if needed
Are all data available and do they match the descriptions in the paper? YesAdditional CommentsNCBI numbers should be added when available.Are the data and metadata consistent with relevant minimum information or reporting standards? See GigaDB checklists for examples <a href="http://gigadb.org/site/guide" target="_blank">http://gigadb.org/site/guide</a>YesAdditional CommentsIs the data acquisition clear, complete and methodologically sound?YesAdditional CommentsIs there sufficient detail in the methods and data-processing steps to allow reproduction?YesAdditional CommentsIs there sufficient data validation and statistical analyses of data quality? YesAdditional CommentsIs the validation suitable for this type of data?YesAdditional CommentsIs there sufficient information for others to reuse this dataset or integrate it with other data?YesAdditional CommentsAny Additional Overall Comments to the AuthorI would recommend using a lower mutation rate for seals: de novo mutation rates around 7e-9 have been measured for a few pinniped species. Line 129: I think you mean kya here (not Ma). Line 160: I think this should be "an average value of 0.066%"RecommendationMinor Revision